# Vertical distribution of planktic foraminifera through an Oxygen Minimum Zone:

## how assemblages and shell morphology reflect oxygen concentrations

Catherine V. Davis[1], Karen Wishner[2], Willem Renema[3] & Pincelli M. Hull[1,4]

[1]Department of Earth and Planetary Sciences, Yale University, New Haven CT 06511,

USA

[2]Graduate School of Oceanography, University of Rhode Island, Narragansett, RI 02882,

USA

[3]Naturalis Biodiversity Center, Leiden, the Netherlands

[4]Peabody Museum of Natural History, Yale University, New Haven 06511, USA

## Abstract

Oxygen-depleted regions of the global ocean are rapidly expanding, with important

implications for global biogeochemical cycles. However, our ability to make projections

of a future deoxygenated ocean is limited by a lack of empirical data with which to test

and constrain the behavior of global climatic and oceanographic models. We use depth-

stratified plankton tows to demonstrate that some species of planktic foraminifera are

adapted to life in the heart of the pelagic Oxygen Minimum Zone (OMZ). In particular,

we identify two species, *Globorotaloides hexagonus* and *Hastigerina parapelagica*,

living within the Eastern Tropical North Pacific OMZ. The shells of the former are

preserved in marine sediments and could be used to trace the extent and intensity of low-

oxygen pelagic habitats in the fossil record. Additional morphometric analyses of *G.*



*hexagonus* show that shells found in the lowest oxygen environments are larger, more

porous, less dense, and have more chambers in the final whorl. The association of this

species with the OMZ and the apparent plasticity of its shell in response to ambient

oxygenation invites the use of *G. hexagonus* shells in sediment cores as potential proxies

for both the presence and intensity of overlying OMZs.

**1. Introduction**

Oxygenation in the oceans is temporally and spatially variable, and is controlled by

physical factors like ventilation, and biotic factors such as photosynthesis and respiration.

Oxygen Minimum Zones (OMZs), where dissolved oxygen can reach undetectable levels,

are found in mid-waters (i.e., water depths of 100s to 1000s of meters) in some regions of

the global ocean. They are often associated with Eastern Boundary Currents, and other

upwelling regions, where surface productivity, and thus sub-surface respiration, is high

and ventilation of intermediate waters is low. The presence and extent of dysoxic and

anoxic waters and ecosystems have an outsized influence on global biogeochemical

cycling (Gruber et al., 2008; DeVries et al., 2012; Breitburg et al., 2018), making the

ongoing expansion and intensification of OMZs (Stramma et al., 2008; Keeling et al.,

2009; Stramma et al., 2010; Levin, 2017; Breitburg et al., 2108) of critical importance to

future ocean health. Despite this, there are limited geologic records with which to

constrain long-term change in pelagic OMZ environments and, consequently,

considerable uncertainty in projections of future OMZs (Stramma et al., 2012; Levin,

45    2017).



Existing tools for detecting the presence and intensity of OMZs on geological time scales have severe limitations. Proxies for marine oxygenation currently fall into three broad categories: 1) those that are indicative of productivity, nutrient utilization, and preservation, such as carbon accumulation and stable isotopes of carbon and nitrogen; 2)

benthic faunal assemblages; and 3) sedimentary indicators such as laminations or accumulation of redox-sensitive trace elements in sediments. Proxies of the first type are indirect indicators of OMZs and cannot deconvolve oxygenation and productivity. Although OMZs are generally associated with highly productive environments today, the formation of an OMZ reflects a combination of factors including source water

oxygenation and local processes like nutrient cycling, primary productivity, and organic matter sinking and degradation rates. Proxies of the second and third types function only when a zone of low oxygen intersects the seafloor, which presents a significant geographic limitation. Thus, there is a real need for the development and application of new environmental and oxygenation proxies for OMZs in order to enhance the

paleoceanographic toolkit for understanding long-term change in these critical environments.

The shells of planktic foraminifera form the basis of some of the most widely used paleoceanographic proxies for reconstructing past pelagic and near-surface environments (see Kucera, 2007; Katz et al., 2010 for reviews). Here we explore the potential of

planktic foraminifera as proxies for the extent and intensity of OMZ environments. Several lines of evidence suggest that planktic foraminifera may occur in low oxygen environments. Laboratory experiments with the species *Orbulina universa* and *Globigerina bulloides* show that both can survive and calcify under low oxygen



conditions (Kuroyanagi et al., 2013), despite living in the ocean mixed layer (e.g.,

Emiliani, 1954; Fairbanks et al., 1982; Field, 2004; Birch et al., 2013; Wejnert et al.,

2013) where they are unlikely to experience sustained low oxygen. Moreover, multiple

species have been hypothesized as low oxygen specialists: the rarely fossilized species,

*Hastigerina digitata,* has been observed *in situ* within low oxygen waters (Hull et al.,

2011), *Globorotaloides hexagonus* has been collected in plankton tows associated with

low oxygen water masses (Ortiz et al., 1995; Birch et al., 2013), and numerous digitate

foraminifers are associated with low oxygen waters in the fossil record (Coxall et al.,

2007). However, without a systematic understanding of species distributions relative to

the OMZ, foraminifera-based oxygen proxies can be interpreted only as reflecting a

general "sub-surface" environment.

OMZs are home to specialized groups of organisms capable of tolerating

extraordinarily low dissolved oxygen levels. A growing body of literature has focused on

the distributions of larger zooplankton (e.g., Wishner et al., 1995; Wishner et al., 1998;

Escribano et al., 2009; Wishner et al., 2013; Maas, et al., 2014; Wishner et al., 2018;

2020a), microbial (e.g., Duret et al., 2015; Podlaska et al., 2012; Medina Faull et al.

2020), and viral (Cassman, et al., 2012) populations that live and cycle nutrients within

the OMZ, but no equivalent study has targeted planktic foraminifera. However, benthic

foraminifera are widely understood to be among the extremophiles that thrive in the

OMZ through special adaptations (Levin, 2003; Bernhard and Bowser, 2008; Glock et al.,

2012; LeKieffre et al., 2017; Gooday, et al. 2020). There they are important contributors

to benthic food webs (e.g., Nomaki et al., 2008; Enge et al., 2014), and are used as



indicators of low-oxygen environments (e.g., Kaiho, 1994; Bernhard et al., 1997; Cannariato et al., 1999; Jorissen et al., 2007; Ohkushi et al., 2013).

The goals of this study are to describe and quantify the abundance of living planktic foraminifera above and within a modern OMZ, to test:

1) whether modern planktic foraminifera are present within the OMZ;

2) whether specific species are preferentially or exclusively living within the OMZ; and

3) whether morphological traits of OMZ-dwelling foraminifera reflect oxygenation levels in the environments from which they are recovered

1.1. The Eastern Tropical North Pacific Oxygen Minimum Zone

The Eastern Tropical Pacific is home to the world's largest OMZ, fueled by a combination of high coastal and equatorial productivity and poorly ventilated sub-thermocline waters (Paulmier and Ruiz-Pino, 2009; Fiedler and Talley, 2006). The OMZ in the Eastern Tropical North Pacific (ETNP) is associated with both a deep particle

maximum and a secondary nitrite maximum, indicative of reduction of nitrate to nitrite within the OMZ (Garfield et al., 1983; Buchwald, et al., 2015; Medina Faull et al., 2020). The region sampled here is located west of the Baja peninsula and removed from the regions of greatest surface productivity, towards the northern reches of the low oxygen tongue of the ETNP OMZ (Fig. 1; Supplemental Fig. 1).


## 2.    Methods

2.1 Plankton Tow Collections



Day and night vertically stratified and horizontal MOCNESS (Multiple

Opening/Closing Net and Environmental Sensing System) tows were taken onboard the

*R/V Sikuliaq.* A 1 m$^2$ updated MOCNESS system with eight or nine 222 μm mesh nets

and a Sea-Bird SBE911 CTD with updated software in place of the original sensors was

used (see Wishner et al., 2018). All tows were carried out within relatively close

proximity to one another (21° N, 117° W) between January 26[th] and February 7[th] 2017

(Wishner et al, 2018, 2020a, 2020b). This study utilized a total of 8 tows, with each tow

including the deployment of 8 to 9 nets to sample a defined depth interval. We use six

depth-stratified vertical profiles (#716, #718, #720, #721, #722, #725) that sampled

portions of the 0 – 1000 m water column, and two horizontal tows that sampled the OMZ

at ~425 m depth (#724, #726) (Wishner et al. 2018, 2020a, 2020b). Vertical strata

sampled by each net were 25 m to 200 m thick, depending on the tow and depth (see net

strata depths and volume filtered for each net in Wishner et al. 2019, 2020b). In

horizontal tows, each net sampled a distance of about 1 km (Wishner et al. 2018).

Environmental data were collected with the MOCNESS CTD sensors simultaneous with

plankton collections. For oxygen, a Sea-Bird SBE43 sensor was used. All plankton

samples were stored in sodium borate-buffered seawater and formalin at sea. Isolation of

foraminifera from samples occurred in 2017-2019 at the University of Rhode Island.

Between 3/10[ths] and 1/125[ths] of material in a sample was examined depending upon

abundance of foraminifera, and all intact shells were isolated from the split.

Foraminifera were identified to the species level by light microscope at the University

of South Carolina and Yale University. Some shells (9% of the total observed) were

either damaged or, more rarely, appeared to be juvenile forms, such that no species-level





identification could be assigned. Due to excellent tissue preservation, the presence or

absence of foraminiferal cytoplasm was identifiable, and foraminifera were classified as

either "live," based on the presence of cytoplasm, or "dead" in the absence of cytoplasm

(Fig. 2). Although, preservation was excellent in most tows, some dissolution was

observed in 3 shallow (< 100 m) nets. These have been excluded from further analyses, to

prevent skewing assemblages towards more dissolution-resistant taxa. We note that these

3 nets were exceptionally high in organic matter and that organic matter degradation was

the likely cause of dissolution despite buffering and a relatively short storage interval.

The organic matter concentration and preservation concerns in these 3 nets do not apply

to the other nets considered in this study.

2.2 Counting and Statistics

        Total counts of foraminifera were adjusted for both the tow split analyzed as well

as the total water volume filtered and are presented as individuals m$^{-3}$ or as relative

abundance. Diversity was calculated using the 'diversity' function and Shannon index in

the R 'vegan' package (Oksanen, et al., 2013). All other statistics were carried out in the

base package in R (R Core Team, 2017).

2.3 Morphological Analyses

All individuals of the species *G. hexagonus* were weighed on a Mettler Toledo

ultramicrobalance (± 1 µg) in the Yale Analytical and Stable Isotope Center and imaged

on a Leica DM6000 light microscope at Yale University. Measurements were made in

ImageJ by identifying a flat section of the F or F-1 chamber minimally affected by glare





and measuring the total area of the section and the total area of section excluding pores.

All other morphometric measurements were made using the AutoMorph software

(Hsiang et al., 2016).

Porosity is reported as the percentage of shell surface area comprised of pores. Size-

normalized weight was assessed by the area density method described by Marshall et al.

(2013), with the weight of each shell normalized to its 2-dimensional surface area. The

compactness of shells was assessed as the ratio of the 2-dimensional surface area to the

area of a circle (the most compact possible geometry) of the same perimeter. The aspect

ratio was defined as the ratio between the height (longest dimension) and width

(perpendicular to the longest dimension) as measured in the AutoMorph software (Hsiang

et al., 2016). Shell size was ascertained by length, surface area, and shell perimeter. As

surface area and shell perimeter were used in deriving compactness and size-normalized

weights, respectively, and all parameters are interrelated, we refer to the longest shell

dimension when referring to size.

Micro CT-scans were generated at the Naturalis Biodiversity Center using a Zeiss

Xradia 520 Versa micro-CT scanner aiming at a voxel size of 0.627 μm; realized

resolution varied from 0.4-0.7 μm. Scans were made at 90 kV using 20X optical

magnification, and were reconstructed using the Zeiss software. Micro CT scans were

processed and analysed in VG Studio, with volumes assessed by creating a mesh wrap in

the MeshLab software (Cignoni et al., 2008) as described in Burke et al. (2020).

**3. Results**

3.1 Hydrological Data from Tows





Plankton tows sampled depths between 0 and 1000 m, across dissolved oxygen levels between 0.03 and 4.93 ml $L^{-1}$ and temperatures ranging from 4.5 to 22.9° C. Although small-scale oxygen features and their depth relative to the oxycline and OMZ varied

somewhat (Wishner et al., 2018, 2020b), the overall hydrographic structure of the water column was consistent across tows. A warm, oxygenated surface mixed layer overlaid an extremely oxygen depleted OMZ, with gradual cooling at increasing depth below the thermocline. The upper oxycline (the zone of rapidly decreasing oxygen) was located between 150 and 250 m water depth, with its upper boundary at the thermocline (Fig. 3-

5). Categorization of oxygen levels follows the discussion of Hofmann et al. (2011) and Moffitt et al. (2015). We defined environments with $[O_2] > 2.45$ ml $L^{-1}$ (109 μM) as oxic, between 2.45 ml $L^{-1}$ and 1.4 ml $L^{-1}$ (63 μM) as transitional ("mild hypoxia" in previous literature), and < 1.4 ml $L^{-1}$ as OMZ conditions ("hypoxia" and below). Previous authors have distinguished between intermediate (0.5-1.4 ml $L^{-1}$) and severe hypoxia (< 0.5 ml $L^{-1}$;

22 μM), but we have collapsed these to 'hypoxia' as foraminiferal assemblages did not differ between the two categories (see Supplemental Table 1).

3.2 Live Foraminiferal Assemblages

Assemblages of live foraminifera, described using the definitions of oxygen outlined

above, can be divided into three categories: those living in oxic conditions (minimum $[O_2]$ within a net > 2.45 ml $L^{-1}$), OMZ conditions (maximum $[O_2]$ within a net < 1.4 ml $L^{-1}$), and transitional (nets sampling between these two extremes). The oxic group was the shallowest, with the deepest tow included in this category extending to only 150 m water depth. These tows had the densest population of foraminifera with 3.4 individuals $m^{-3}$ and



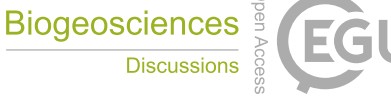

the greatest diversity with a mean Shannon index value of 1.3 (ranging from 1.2 to 1.5

across 5 nets). In this relatively shallow, oxic environment, the assemblage was

dominated by *Trilobatus sacculifer* (74.6%) followed by *Globigerinoides ruber* (5.4%),

*Hastigerina pelagica* (5.0%), *Globigerinella siphonifera* (4.0%), *Orbulina universa*

(3.5%), *Globorotaloides hexagonus* (3.1%), and *Globigerina bulloides* (1.9%). The

species *Hastigerina parapelagica*, *Globorotalia menardii*, *Globoquadrina conglomerata,*

*Pulleniatina obliquiloculata*, and *Globorotalia tumida* were all found in low abundance

(<1%). *Globigerinita glutinata, Hastigerina digitata, Globigerinoides conglobatus*,

*Sphaeroidinella dehiscens*, *Candeina nitida*, *Globigerinella calida*, *Globigerina*

*falconensis*, and *Neogloboquadrina dutertrei* were absent (Table 1; Fig. 6).

Foraminifera from the OMZ assemblage were found in nets collected below 250 m

water depth, and occurred at much lower densities of 0.2 individuals m$^{-3}$. This

assemblage was heavily dominated by *G. hexagonus* (86.1%), followed by *G. sacculifer*

(3.6%), *H. parapelagica* (2.0%), *H. pelagica* (1.4%), and *G. menardii* (0.8%). The

species *G. ruber*, *O. universa*, *G. siphonifera, G. glutinata, G. conglobatus, P.*

*obliquiloculata,* and *G. bulloides* were found in low abundance (<1%), and all other

species were absent (Table 1; Fig. 6). The OMZ assemblages were also the least diverse,

with a mean Shannon index value of 0.9 (ranging from 0.8 to 1.0 in 54 nets).

        The transitional assemblages primarily represented depths between 100 and 250 m

and had the lowest density of foraminifera with 0.1 individuals m$^{-3}$. There was one net

that sampled 800 to 1000 m and would also fall into this oxygen categorization, but was

excluded from analyses as it contained only a few *G. ruber* (< 0.01 individuals m$^{-3}$)

which were likely dead, and is not readily comparable to the upper oxycline habitat of



other transitional samples. The transitional assemblage was nearly as diverse as the oxic

assemblage with a Shannon index of 1.2 (ranging from 1.1 to 1.2 across 4 nets). It was

composed of *G. hexagonus* (40.7%), *G. sacculifer* (22.1%), *G. siphonifera* (9.6%), *G.*

*conglomerata* (6.4%), *O. universa* (5.5%), *Globorotalia menardii* (5.0%), *H. pelagica*

(3.9%), *G. conglobatus* (2.4%)*,* and *S. dehiscens* (1.6%). A few other species, *H.*

*parapelagica*, *C. nitida*, *G. ruber*, and *P. obliquiloculata*, were found in abundances <

1% (Table 1; Fig. 6).


### 3.3 Dead Foraminiferal Assemblages

Dead assemblages mirrored living assemblages, with high species diversity (Shannon

index > 1) in depths up to 400 m, after which diversity declined to Shannon index values

between 0.5 and 1. An average of 0.2 empty shells $m^{-3}$ were recovered for all tows. The

majority of the dead assemblage was made up of *G. sacculifer* (55.4%), followed by *H.*

*pelagica* (11.7%), *G. ruber* (6.4%), *G. siphonifera* (6.0%), *G. hexagonus* (5.9%), and *O.*

*universa* (5.1%). All other species comprised less than 2% of the assemblage. While

every species occurring with cytoplasm was also found without cytoplasm, two species,

*Hastigerina digitata* and *Neogloboquadrina dutertrei,* were identified in low abundances

without cytoplasm, but were not observed with cytoplasm.

### 3.4 Morphological Variation in *G. hexagonus*

### 3.4.1 Porosity

Porosity of the final chamber in *G. hexagonus* was highly variable among

individuals and among tows, ranging from 1.7% to 19.4% of the surface area measured





by light microscope. Porosity decreased as oxygen increased, with the clearest

relationship between the log of porosity and log of dissolved oxygen ($R^2 = 0.38$, p-value

< 0.001). A comparison between porosity of the final chamber, measured by CT scan and

light microscope, showed that CT measurements consistently demonstrated higher

porosities (Fig. 7). This methodology allowed for non-destructive imaging of the inner

shell unobscured by later calcite growth, as well as higher resolution, and should be

considered a more accurate measure of shell porosity. A comparison of the two methods

carried out on a subset of shells (n = 31) showed that the results from the two approaches

are correlated ($R^2 = 0.37$, p-value < 0.001; Fig. 7), indicating that the less labor-intensive

use of light-microscope measurements captures some of the same trend as the CT-based

approach. Final chamber porosity increased linearly with the size across individuals ($R^2 =$

0.33, p-value < 0.001), and with ontogeny within individuals (Fig. 8), demonstrating an

interaction between size, ontogeny, and porosity.

3.4.2   Size and Chamber Number

Size decreased with the log of oxygen (Spearman's $\rho$ = -0.64; p-value < 0.001).

The largest change in size, as well as the largest change in size-normalized weight and

chamber number, was a step change corresponding to oxygen levels between 0.1 and 0.2

ml $L^{-1}$ (Fig. 9). The number of chambers visible in the final whorl ranged between 4 and

7 (net means between 4.8 and 6.1) and the largest change in mean chamber number also

occurred between 0.1 and 0.2 ml $L^{-1}$ $O_2$, with shells having a greater number of chambers

in the final whorl in low oxygen tows (correlation of chamber number to log of average

oxygen: Spearman's $\rho$ = -0.68; p-value < 0.001; Fig. 9).





### 3.4.3 Size-normalized Weight

*Globorotaloides hexagonus* shells were light for their size, with individual shell weights averaging just 7.7 µg, and ranging from 1 to 22 µg for shells sized between 297 and 631 µm in length. Size-normalized weight increased with oxygenation, especially below 0.2 ml L$^{-1}$ O$_2$ (correlation of size-normalized weight to the log of oxygen: Spearman's $\rho = 0.52$; p-value < 0.001; Fig. 9). Size-normalized weight and porosity were correlated ($R^2 = 0.34$; p-value < 0.001), as were calcite volume and final chamber porosity measured in CT-scanned foraminifera ($R^2 = 0.18$; p-value < 0.001; Supplemental Fig. 2). Size-normalized weight is also dependent upon size (Henehan et al., 2017), although in our study the variance in size-normalized weight explained by size was low ($R^2 = 0.10$ p-value < 0.001).

### 3.4.4 Compactness and Aspect Ratio

We further tested the utility of shell compactness and aspect ratios as potentially diagnostic of the morphological gradient observed. Although shell compactness increased linearly with oxygenation ($R^2 = 0.03$ p-value = 0.04) and aspect ratio decreased linearly with the log of oxygen ($R^2 = 0.09$ p-value < 0.001), oxygenation accounted for very little of the variance in either parameter and they were not considered further.

## 4. Discussion

### 4.1 Distinct OMZ Community of Planktic Foraminifera



Live foraminifera obtained from vertical profiles with depth-stratified nets in the

ETNP form three distinct pelagic assemblages associated with differing oxygen levels.

The OMZ community, living at the lowest oxygen level, was typified by the presence and

high relative abundance of the foraminifer *G. hexagonus*. This species can be considered

an indicator of an OMZ habitat and may be useful as an OMZ marker in sedimentary

records, as discussed below.

The shallow, oxic assemblage (< 150 m) of planktic foraminifera was relatively

diverse and included species typical of the Pacific subtropical gyre (Eguchi et al., 1999;

Kuroyanagi et al., 2002), with affinities for warmer sea surface temperatures and

oligotrophic conditions. However, there was substantial variation between the three tows

for which surface assemblages were available (#716, #721, and #725), with abundances

in the upper 100 m varying from < 0.1 individuals $m^{-3}$ (tow #716) to 3.0 individuals $m^{-3}$

(tow #721) and 11.0 individuals $m^{-3}$ (tow #725) (Fig. 3-5). In the latter two tows the

majority of the assemblage was comprised of *T. sacculifer*, whereas in Tow #716, *G.*

*menardii* was the most abundant species comprising 33% of the total assemblage. A

slightly shallower thermocline (compare Fig. 3 to Fig. 4 and 5) and deep chlorophyll

maximum may be partially responsible for differing abundances. However, there may

also be a lunar-associated reproductive response affecting abundance patterns. Tow #716

was taken during a waning moon, but tows #721 and #725 were taken during a waxing

moon (USNO, accessed 10/10/2019). *Trilobatus sacculifer* reproduces on a lunar cycle,

with the largest sizes reached just prior to reproduction during the full moon (Bijma et al.,

1990; Erez et al., 1991; Kawahata et al., 2002; Lin et al., 2010; Jonkers et al., 2015;

Venancio et al., 2016). As a result, more large individuals of this species were likely to be



present and thus sampled in our 222 µm mesh nets prior to a full moon (tows #721 and

#725).

The OMZ assemblage was dominated by the species *G. hexagonus*, followed by *T. sacculifer* and *H. parapelagica*. Use of presence/absence of cytoplasm as an indicator for living foraminifera results in an overestimation of live individuals, as dead individuals may retain some cytoplasm while live individuals cannot be devoid of cytoplasm. Thus,

despite the presence of *T. sacculifer* in several OMZ samples, it is unlikely that this species, which has photosymbionts and a relatively shallow, photic zone habitat (Fairbanks et al., 1982; Ravelo & Fairbanks, 1992; Regenberg, et al., 2009; Birch et al., 2013; Rebotim et al., 2017), was resident in the deep OMZ. It is more likely that cytoplasm-bearing shells of *T. sacculifer* found below the photic zone are a consequence

of their very high abundance in the surface ocean and reflected premature mortality and/or the retention of cytoplasm following reproduction. On the other hand, *G. hexagonus* and *H. parapelagica* comprised 88.1% of cytoplasm-bearing shells in OMZ nets, while only found in low abundances in surface assemblages. This suggests that these two species are truly endemic to deeper hypoxic waters.

The transitional assemblage was a mix between the well-oxygenated surface assemblage, with abundant *T. sacculifer*, and the deeper OMZ assemblage, composed primarily of *G. hexagonus*. This mix of species was almost certainly an artifact of the depth (and oxygen) range integrated within a single net (50-100 m thick strata) through the steep oxycline. However, the transitional assemblage also had two unique

characteristics. The first was the presence of deeper-dwelling taxa, such as *G. conglomerata* and *G. menardii*, which were rare in most other nets. The second was the



exceptionally low density of planktic foraminifera (mean of 0.1 individual per m$^{-3}$ across

4 tows; Fig. 3-5). The low density of foraminifera in the oxycline was an interesting

contrast to the vertical distributions of many metazoan species that often peak in

abundance in the upper oxycline and decline in the core of the OMZ (Maas et al. 2014,

Wishner et al., 1995, 2013, 2020b). Based on the mixed assemblage and low densities,

we hypothesize that planktic foraminifera are largely absent from the upper oxycline,

with populations restricted to either the oxygenated photic zone habitat above or the

OMZ below. Future sampling at higher vertical resolution through the oxycline is

required to test this hypothesis. Whether this distributional pattern is related to

physiological constraints, food resources, physical oceanographic mechanisms, or other

environmental parameters is unknown.

### 4.2 *Globorotaloides hexagonus* as an OMZ Indicator Species

*Globorotaloides hexagonus* was consistently found within our low oxygen nets,

though individuals were sparsely distributed (mean density of 0.2 individual m$^{-3}$), with

peak abundances between 300-500 m depth in the core of the OMZ (Fig. 3-5;

Supplemental Fig. 3-5). There was no evidence of diel vertical migration when

comparing distributions in tows taken during the day (#718, #722, #724, #725, #726) and

night (#716, #720, #721), in agreement with the lack of diel vertical migration observed

in shallow-dwelling species (Meilland et al., 2019). Absence of large-scale migrations

and a preference for extremely oxygen-depleted habitats indicate that the species is

adapted to live for long periods of time, likely its entire lifespan, within extremely low

oxygen conditions.



*Globorotaloides hexagonus* has previously been associated with deep, low oxygen

water masses across the Indo-Pacific, including the Eastern North Pacific (Sautter &

Thunell, 1991; Ortiz et al., 1996; Davis et al., 2016), Equatorial Pacific (Fairbanks et al.,

1982; Rippert et al., 2016; Max et al., 2017; Rippert et al., 2017), the Peru-Chile margin

(Marchant et al., 1998) and the Indian Ocean (Rao et al., 1989; Birch et al., 2013).

However, the species is frequently assumed to be extinct in the Atlantic, with recent

identifications of *G. hexagonus* in Atlantic sediments explicitly used to date sediments as

pre-Holocene or ascribed to taxonomic error (e.g., Kucera et al., 2005; Siccha & Kucera,

2017). The assumption of a basin-wide extinction appears poorly supported, and *G.*

*hexagonus* shells were isolated from deep (500 – 3200 m) Atlantic sediment traps as

recently as 2009-2013 (Smart et al., 2018). We hypothesize that *G. hexagonus* occupies

low-oxygen mid-waters globally (i.e., in the Atlantic as well as the Indo-Pacific), but that

its deep habitat, low abundance, and the historical dearth of surveys of living planktic

foraminifera in low $O_2$ regions along the western African margin have biased

observations of *G. hexagonus* in the modern Atlantic. Altogether, the geographic

distribution, presence of cytoplasm-bearing *G. hexagonus* in OMZ tows, and scarcity of

*G. hexagonus* above the oxycline, strongly suggest that *G. hexagonus* lives preferentially,

or even exclusively, within the OMZ.

      We also found a second, less abundant, species, *H. parapelagica*, in association

with low oxygen waters. This same morphology was previously observed *in situ* in low

oxygen waters by Hull et al. (2011), and more recently by Gaskell et al. (2019), referred

to as "*Hastigerina* spp." by the former and "*Hastigerina pelagica*" by the latter. Given

the depth distribution and morphological variation observed here for *H. parapelagica,* we





suspect that it is synonymous with the globally distributed "*Hastigerina pelagica*"

genotype IIa, described by Weiner et al. (2012) and use the name *Hastigerina*

*parapelagica* (Saito et al., 1976) as the senior synonym of *Hastigerina pelagica* genotype

IIa (Weiner et al. 2012).

### 4.3  Morphological Variation in *G. hexagonus* Reflects Water Column Oxygenation

*Globorotaloides hexagonus* shares several morphological traits with low-oxygen

associated benthic foraminifera including a flattened whorl maximizing its surface

area/volume ratio at a given size and large pores (e.g., Bernhard, 1986). Both characters

could serve to increase gas exchange and fulfill metabolic requirements in an oxygen-

limited environment (Leutenegger & Hansen, 1979; Corliss, 1985). Unlike some digitate

planktic foraminifera previously associated with deep and oxygen depleted environments

(Hull et al., 2011; Coxall et al., 2007; Gaskell et al., 2019), *G. hexagonus* is non-spinose,

which may suggest that it is herbivorous or bacterivorous as described for other non-

spinose foraminifera (Schiebel & Hemleben 2017; Bird et al., 2018), rather than

dependent on live zooplankton as prey.

The shells of *G. hexagonus* in deeper, less oxygenated waters appeared more

porous, larger, and less compact than those from shallower, more oxygenated

environments. These observations, and the presence of *G. hexagonus* across a wide range

of depths and oxygenation levels, led us to quantify the environmental correlates of

morphological variation in porosity, size-normalized weight, size, chamber number and

shape as potential proxies in paleo-environmental reconstructions. A high shell porosity

and high pore density have been widely associated with low oxygen environments in



benthic foraminifera (Bernhard, 1986; Perez-Cruz & Machain-Castillo, 1990; Glock et

al., 2011, 2012; Kuhnt et al., 2013, 2014; Rathburn et al., 2018) and in cultured planktic

foraminifera (Kuroyanagi et al., 2013). These characteristics may play an important role

in facilitating gas exchange (Leutenegger & Hansen, 1979; Corliss, 1985). However,

increased porosity has also been associated with other parameters: increasing temperature

(Bijma et al., 1990; Burke et al., 2018), decreasing nitrate availability (Glock et al.; 2011,

2018), and increasing shell size (Burke et al., 2018). In the OMZ samples where *G.*

*hexagonus* was found, porosity increased with both decreasing oxygen concentration and

increasing shell size, with the lowest oxygen conditions hosting the largest and most

porous shells (Fig. 9). In contrast to this trend, porosity decreases through ontogeny in *G.*

*hexagonus* with the most recent chamber being less porous than earlier chambers (Fig. 8).

Neither temperature nor nitrate availability (used by some benthic foraminifera as an

alternative terminal proton acceptor in very low oxygen environments; Risgaard-Petersen

et al., 2006; Hogsland et al., 2008; Pina-Ochoa et al., 2010; Bernhard et al., 2011, 2012a,

2012b; Woehle et al., 2018), are likely to drive the observed variation in porosity as

temperature was nearly constant (7.7-8.5 °C) across samples and nitrate availability

increased with depth (Podlaska et al., 2012; Buchwald et al., 2105; Medina Faull et al.,

2020).

Shells collected at lower oxygen levels also had lower size-normalized weights, a

property which negatively correlates with porosity. Size-normalized weight in planktic

foraminifera has frequently been associated with changes in carbonate chemistry (i.e.,

Bijma et al., 2002; Russell et al., 2004; Marshall et al., 2013). As oxygen and DIC depth

profiles in the ocean are inversely related, the OMZ is also a region of exceptionally high





DIC (Paulmier et al., 2008, 2011). While no carbonate chemistry measurements are

available in conjunction with our tows, calcite saturation state at equivalent latitudes in

the Eastern Tropical South Pacific OMZ approaches 1, below which calcite dissolution is

favored (Bates, 2018). Both an increase in porosity, as well as a decrease in size-

normalized weight (whether due to porosity, a decrease in shell thickness, or a

combination of factors), is consistent with a reduction of overall calcification in low

oxygen, DIC rich environments, where precipitation and maintenance of a shell may be

more metabolically expensive.

        Shells collected from the lowest oxygen conditions tended to be larger and less

compact, with more chambers visible in the final whorl. Both decreased compactness and

the addition of more lobes via increased chamber number have the effect of increasing

the surface area/volume ratio at a given size, which could facilitate increased gas

exchange via diffusion. However, the increase in size with decreased oxygen availability

overwhelmed these other morphological features, such that larger *G. hexagonus* in low

oxygen environments had lower surface/volume ratios than smaller individuals from

more oxygenated environments (Supplemental Fig. 6).

Although the increase in size at low oxygen levels appears enigmatic, there are

several potential reasons that could account for this pattern. One benefit could be a larger

surface area for interception of food. Alternatively, increased size (cell volume) has been

associated with greater capacity for denitrification in some benthic foraminifera (Glock et

al., 2019). An inconsistent relationship between surface area/volume ratios and

oxygenation has also been observed in several facultative anaerobic species of benthic

foraminifera (Keating-Bitonti and Payne, 2017). Whether *G. hexagonus* possesses





physiological strategies that allow it to function as a facultative anaerobe cannot be
determined at this point. However, the combination of increased size (potentially
indicative of anaerobic strategies) and increased porosity and morphologies apparently

optimized for increasing aerobic capacity in low oxygen environments, suggest a
complex physiology. A decrease in porosity with ontogeny could even hint at a shift in
physiology over the lifespan of an individual (Fig. 8). Further unraveling the
environmental pressures driving shell morphology in *G. hexagonus* will require a greater
understanding of the species' ecology.


## 5   Conclusions

Vertically-stratified plankton tows taken through the Eastern Tropical North
Pacific show that distinct assemblages of planktic foraminifera live above and within the
OMZ, and that a depauperate fauna occupies the upper oxycline. Two species, *G.*

*hexagonus* and *H. parapelagica,* were found preferentially or exclusively within the
OMZ. Several aspects of shell morphology in *G. hexagonus* varied in response to
ambient oxygen levels. Some morphological features may be associated with facilitating
gas exchange (i.e., porosity, chamber arrangement) or decreasing expenditure on
calcification (size-normalized weight) under the low oxygen and/or carbonate saturation

states of the OMZ. The function of other morphological trends, like size, remain
enigmatic. Abundance patterns and the co-variation of specific morphological features
with oxygenation levels in *G. hexagonus* shells could be used to reconstruct changes in
OMZ environments, providing an additional proxy record of the mid-water OMZ in
which these foraminifera lived. As the species appears to be living primarily or





exclusively in the OMZ, recovery of *G. hexagonus* shells from sediments would be a

strong indication of low-oxygen mid-waters. Moreover, large shells with high porosity,

low size-normalized weight and more chambers in the final whorl could be interpreted as

having calcified closer to the core of the OMZ than their smaller, less porous

conspecifics.


**Acknowledgements**

Many thanks to D. Outram for laboratory assistance and to R.C. Thunell (deceased) for

invaluable support in the genesis of this project. This research was supported by NSF

OCE 1851589 to Davis, NSF OCE 1459243 to Wishner, Seibel, and Roman, and a Sloan

Research Fellowship to P.M.H.







**Figures**

| Species | % of Oxic Assemblage | % of Transitional Assemblage | % of OMZ Assemblage |
|---|---|---|---|
| *T. sacculifer* | 74.6 | 22.1 | 3.6 |
| *G. ruber* | 5.4 | 0.6 | 0.3 |
| *H. pelagica* | 5.0 | 3.9 | 1.4 |
| *G. siphonifera* | 4.0 | 9.6 | 1.0 |
| *O. universa* | 3.5 | 5.5 | 0.1 |
| *G. hexagonus* | 3.1 | 40.7 | 86.1 |
| *G. bulloides* | 1.9 | 0.0 | 0.1 |
| *Hastigerina* spp. | 0.3 | 0.8 | 2.0 |
| *G. menardii* | 0.9 | 5.0 | 0.8 |
| *G. conglomerata* | 1.0 | 6.4 | 0.1 |
| *P. obliquiloculata* | 0.2 | 0.7 | 0.7 |
| *G. tumida* | 0.2 | 0.0 | <0.1 |
| *G. glutinata* | 0.0 | 0.0 | 3.2 |
| *H. digitata* | 0.0 | 0.0 | 0.0 |
| *G. conglobatus* | 0.0 | 2.4 | 0.4 |
| *S. dehiscens* | 0.0 | 1.6 | 0.3 |
| *C. nitida* | 0.0 | 0.7 | 0.0 |
| *G. calida* | 0.0 | 0.0 | <0.1 |
| *G. falconensis* | 0.0 | 0.0 | 0.0 |
| *N. dutertrei* | 0.0 | 0.0 | 0.0 |


**Table 1.** The relative abundance of planktic foraminifera within oxygen defined

assemblages: an oxic assemblage (minimum $O_2$ within a net $O_2$ > 2.45 ml $L^{-1}$),

transitional assemblage, and OMZ assemblage (maximum $O_2$ within a net < 1.4 ml $L^{-1}$).






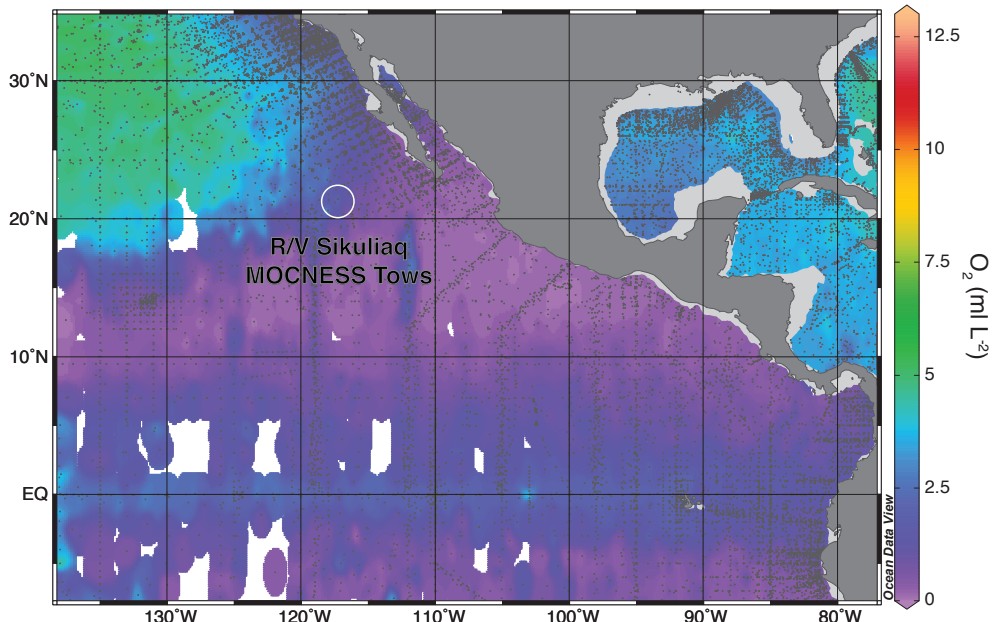

**Fig. 1.** Location of MOCNESS tows (white circle) taken onboard the *R/V Sikuliaq* shown

on a map of dissolved oxygen measured at 200 m below the sea surface. Oxygen data are

aggregated from the World Ocean Atlas (Garcia et al., 2018) and plotted using Ocean

Data Viewer.



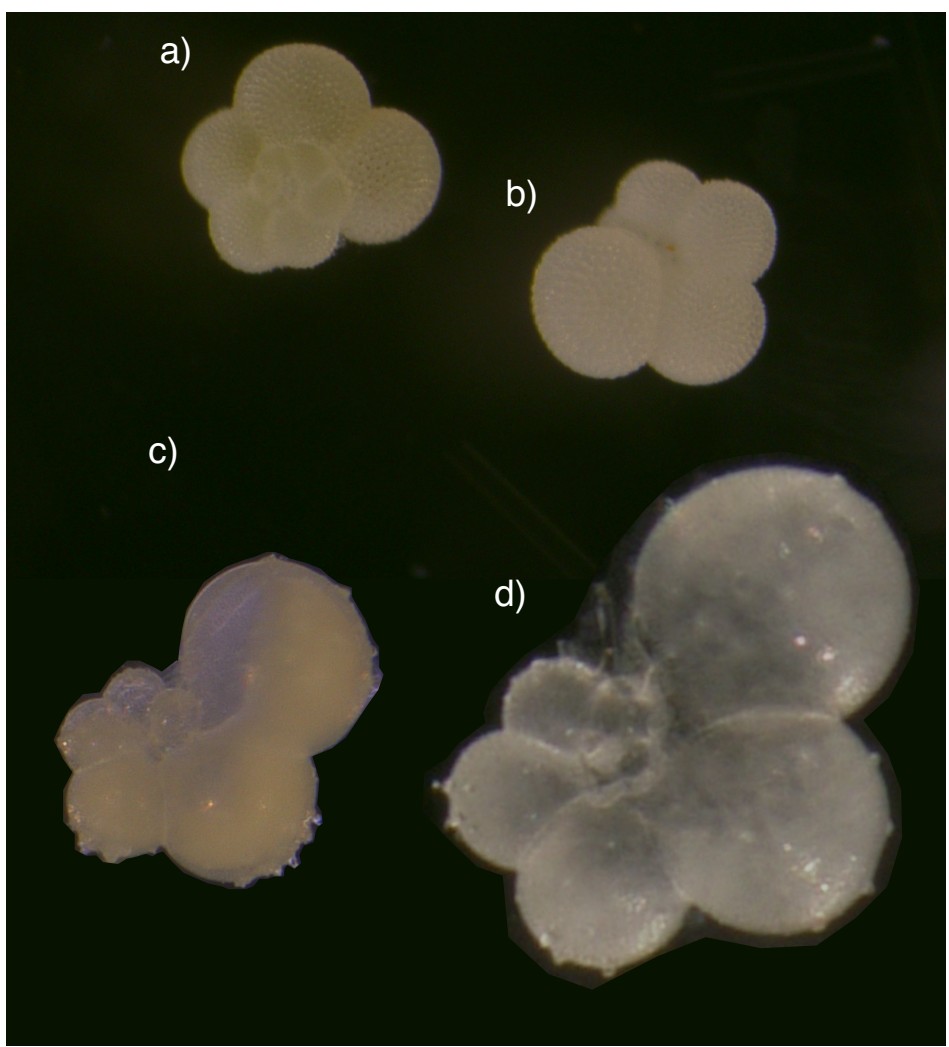

**Fig. 2.** A side-by-side comparison from the same tow of (a) a dorsal view of a live

(cytoplasm containing) *G. hexagonus* and (b) a ventral view of a dead (empty) *G.*

*hexagonus*, as well as (c) a live and (d) dead *H. parapelagica*

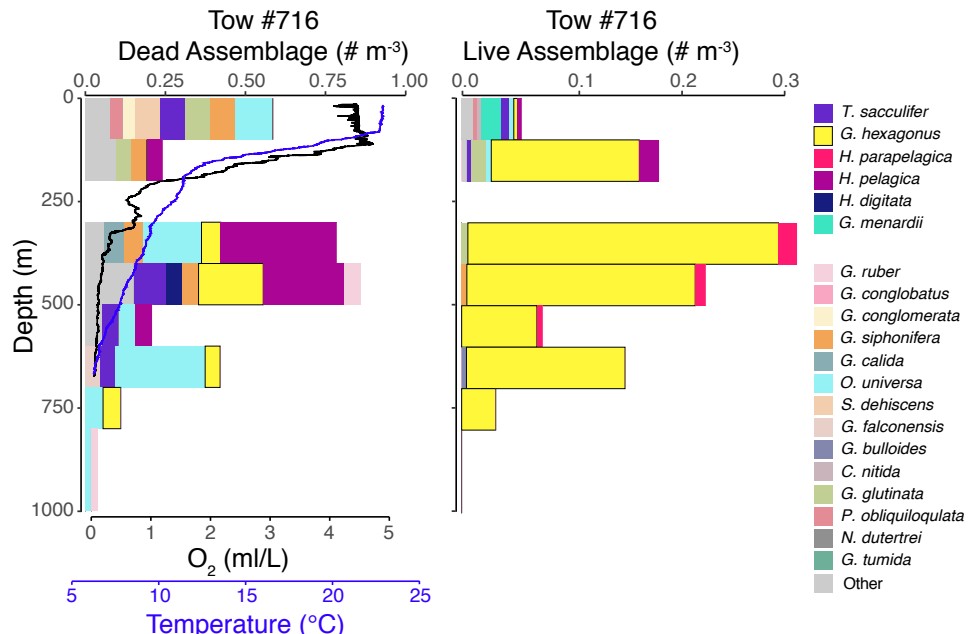

**Fig. 3.** Vertical profiles of the dead foraminiferal assemblage, dissolved oxygen and

temperature (left) and live foraminiferal assemblage (right) from tow #716 (0-1000 m).

Each color represents a different species (see legend), with brighter colors for the six

most salient species across nets and depths. Note that the abundance axes vary between

panels.

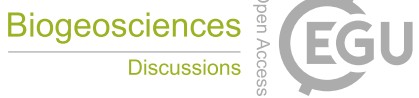

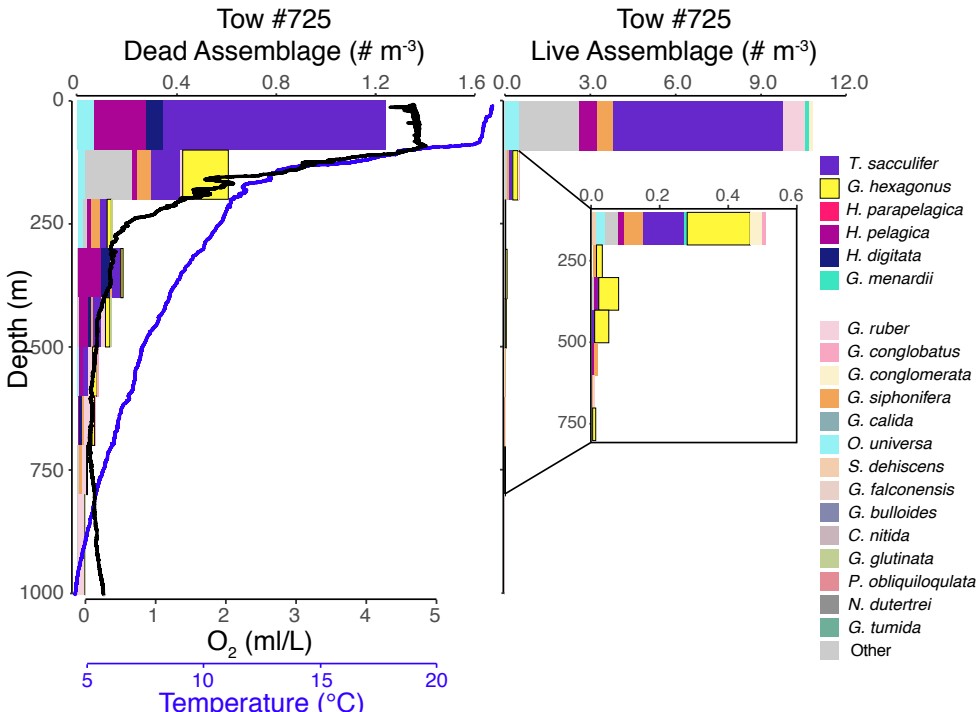

**Fig. 4.** Vertical profiles of the dead foraminiferal assemblage, dissolved oxygen and

temperature (left) and live foraminiferal assemblage (right) from tow #725 (0-1000 m).

Each color represents a different species (see legend). Abundance axes vary, with the

inset showing an enlargement of abundance data in that part of the water column.





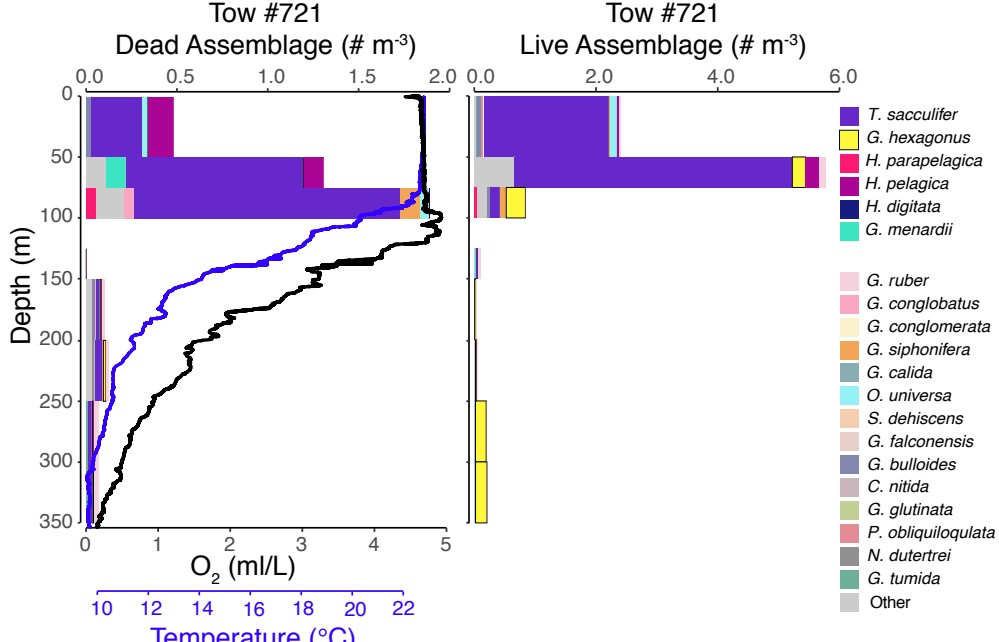

**Fig. 5.** Vertical profiles of the dead foraminiferal assemblage, dissolved oxygen and

temperature (left) and live foraminiferal assemblage (right) from tow #721 (0-350 m).

Each color represents a different species (see legend). Abundance axes vary between

panels.


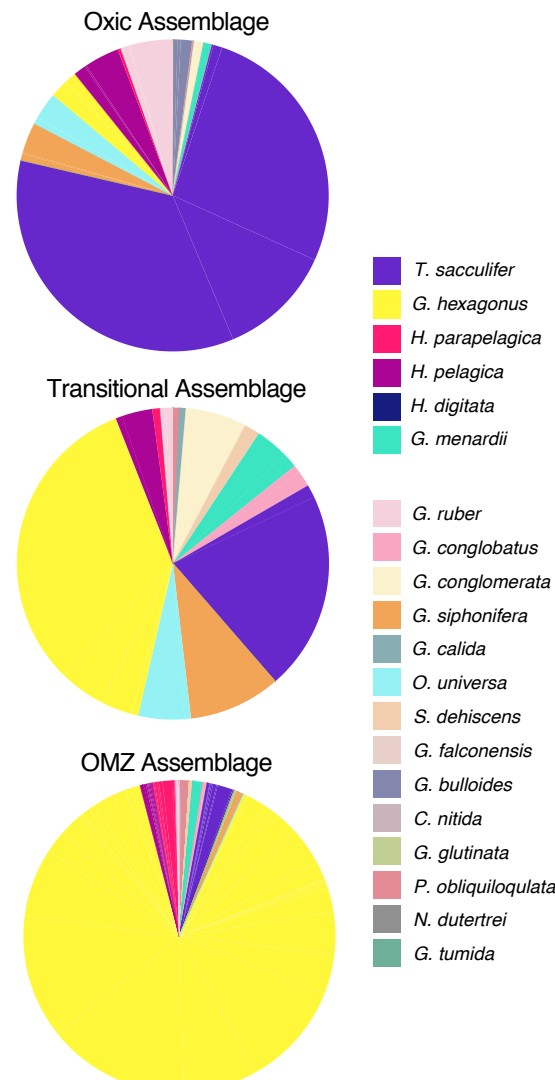

**Fig. 6.** Pie plots of the live foraminiferal assemblages recovered from oxic nets
(minimum dissolved oxygen > 2.45 ml L$^{-1}$; top), transitional nets (middle) and OMZ nets
(maximum dissolved oxygen < 1.4 ml L$^{-1}$; bottom). Each color represents a different
species (see legend).



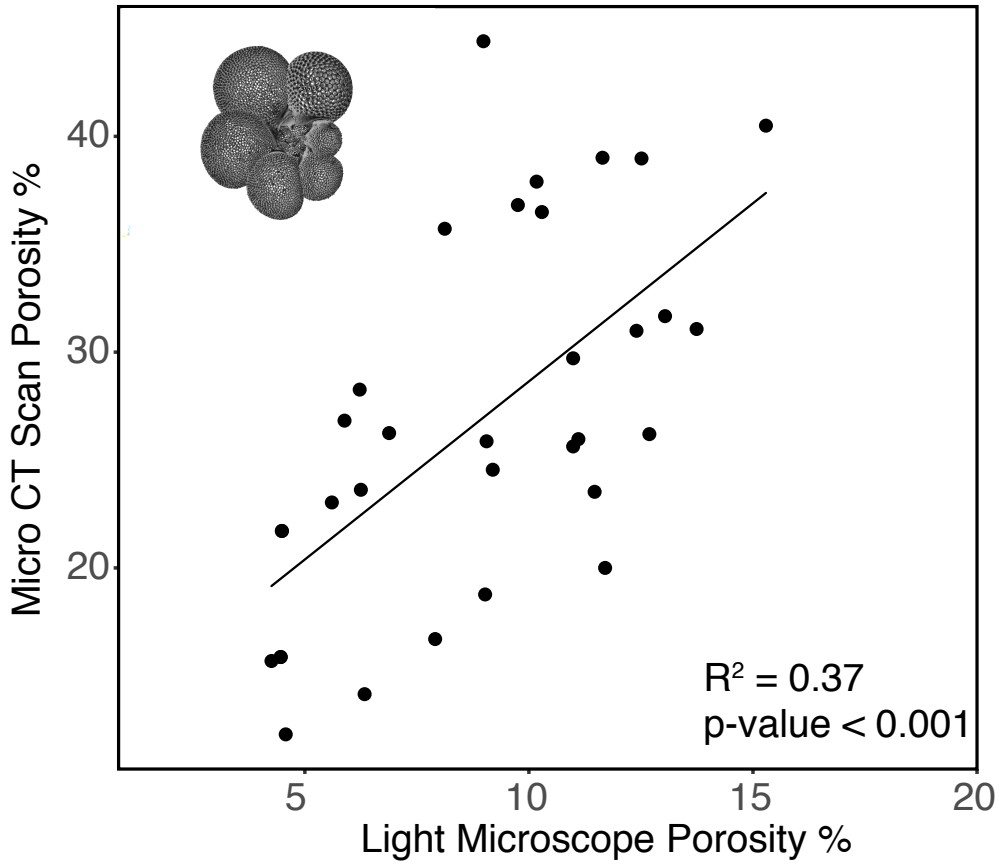

**Figure 7.** Relationship between *G. hexagonus* final chamber porosity measured by light

microscope or CT-scan ($R^2 = 0.45$, p-value < 0.001). A representative image

reconstructed from CT-scanning is inset in the upper left corner.

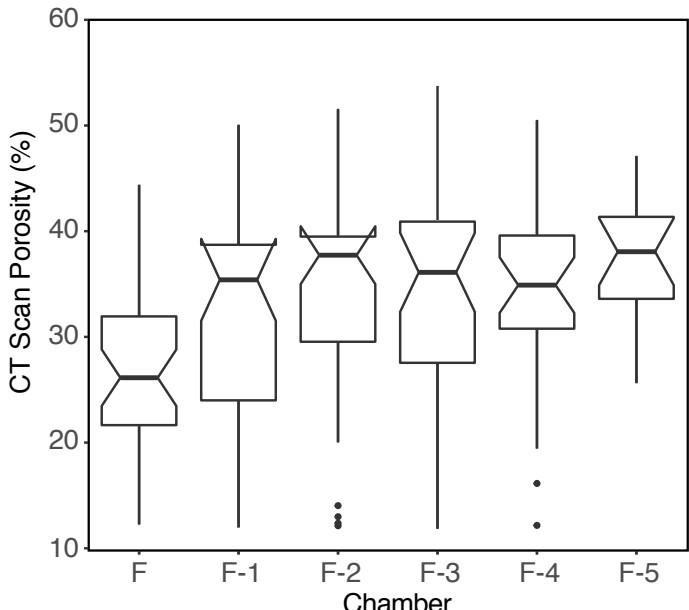

**Fig. 8.** Boxplots of *G. hexagonus* shell porosity, determined by inside-out analyses of CT

scan images, showing an increase in porosity in the most recently formed, F chamber.

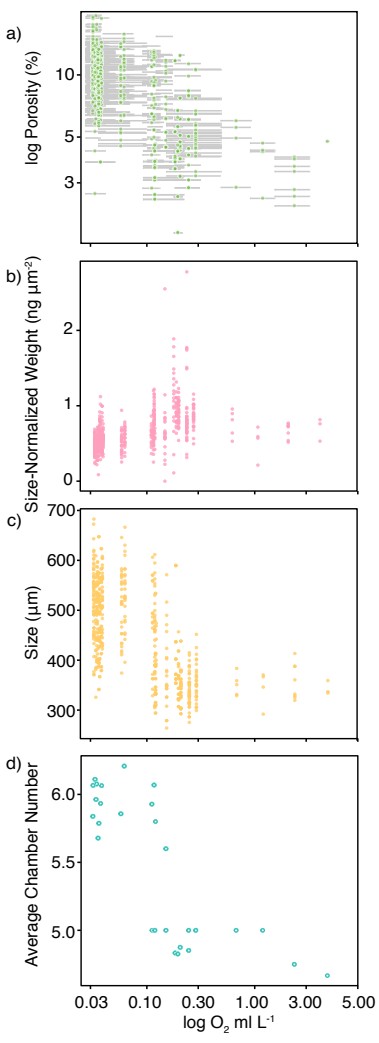


**Fig. 9**. Morphological traits of *G. hexagonus* shells plotted against the average dissolved

oxygen (log scale) measured in the nets in which they were collected. The depicted

characteristics are a) log of porosity, b) size-normalized weight (using the area-density

method), c) size as measured by the longest dimension, and d) the average number of

chambers in the final whorl in a tow. Horizontal gray bars in a) show the range of oxygen

measured for each net.





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
