# Peer review of "Vertical distribution of planktic foraminifera through an Oxygen Minimum Zone: how assemblages and test morphology reflect oxygen concentrations"

_Biogeosciences, 2020_

## Referee Comment (RC1) · Ralf Schiebel (Referee) · 9 Sep 2020

The paper of Catherine Davis and coauthors on "Vertical distribution of planktic foraminifera through an Oxygen Minimum Zone: how assemblages and shell morphology reflect oxygen concentrations" is well written, and adds important information on the ecology and paleoecology of the so far poorly understood planktic foraminifer species Globorotaloides hexagonus. The species has been known to be associated to oxygen minimum zones (OMZs), which are currently expanding as a result of overall rising air and ocean temperatures. Consequently, G. hexagonus may be used as an

indicator of past and future climate change. I would suggest to accept the paper with moderate revisions on the following points.

Most importantly, I would strongly suggest to convey analyses of the molecular genetics of the specimens of the two morphotypes (line 394). For unequivocal proof of the species concept, modern papers of the kind presented here may accept the great opportunity of molecular genetics, and not only rely on the morphospecies concept. Also line 404-406: "The shells of G. hexagonus in deeper, less oxygenated waters appeared more porous, larger, and less compact than those from shallower, more oxygenated environments." These specimens may or may not represent two different genotypes.

Second most importantly, I would strongly suggest to change the statements in lines 375-379, which are not substantiated by data: "We hypothesize that G. hexagonus occupies low-oxygen mid-waters globally (i.e., in the Atlantic as well as the Indo-Pacific), but that its deep habitat, low abundance, and the historical dearth of surveys of living planktic foraminifera in low O2 regions along the western African margin have biased observations of G. hexagonus in the modern Atlantic." – Many studies of sediment trap and net tow samples in the South Atlantic off Namibia (Loncaric and colleagues from Bremen) and the Congo River mouth (Ufkes et al.), as well as surface sediment, would have certainly detected G. hexagonus if present. I have myself seen many net tow, sediment trap, and bottom sediment samples from the Atlantic and other ocean basins, including my PhD project on benthic foraminifers from surface sediments in the Gulf of Guinea, and I have never seen a test of hexagonus in that region.

The following references may be added to support the findings of Davis et al.: Line 40: please refer to the nice paper of Schmidtko et al. 2017 on modern OMZs Line 75: please add also Warren 1994, see Schiebel and Hemleben 2017 Line 88: see also: Glock, N., et al. (2018) Nature Communications, 9, 1217, doi:10.1038/s41467-018-03647-5 Glock, N., et al. (2019) PNAS, 116 (8), 2860-2865, doi:10.1073/pnas.1813887116 Line 327: Please refer to Schiebel et al. (2004) for T.

sacculifer in another study in a region with a prominent subsurface OMZ, i.e. the Arabian Sea. Line 369: refer again to Schiebel et al. (2004)

Some minor issues to be addressed: Line 108: change reches to reaches Line 125: please give the net strata depths and volume filtered for each net in Wishner et al. 2019, 2020b here as well. This does not consume much space, and saves the reader from consulting for additional literature. Lines 125-126: what was the filtered water volume? Line 158: you may want to explain the abbreviations F and F-1 at first mention. Lines 212-214, and 220-221: You may skip the sentence on which species were present and absent. Many other species were possibly absent as well, and are not mentioned. The information on the presence and absence of species should also be available from figures and data tables. Line 227: Foraminifers do not really die in most cases, but reproduce. Therefore, you may change "dead" for "empty" (tests). Line 234: "Empty test assemblages" Lines 299-301: "This species can be considered an indicator of an OMZ habitat and may be useful 300 as an OMZ marker in sedimentary records, as discussed below." This should possibly be the final statement of the section. BTW: This finding is not new to science; please refer to the respective literature. Lines 302-310 present a repetition of the "Results". Please rewrite. Line 318: "larger" reads better than "more large" to me. Lines 322-324: "Use of presence/absence of cytoplasm as an indicator for living foraminifera results in an overestimation of live individuals, as dead individuals may retain some cytoplasm while live individuals cannot be devoid of cytoplasm." This is possibly not entirely true, since decease is most often caused by reproduction, and cytoplasm is consumed and partly converted into offspring (see above). Line 427: Buchmann year of publication Lines 431-432: "(e.g., Bijma. . ." Some species increase in weight, others decrease. Please see Beer et al., Geology, 2010, using samples from the Arabian Sea, i.e. another OMZ region. Line 447: better use outnumber or surpass instead of overwhelm. Line 453: better "as in some benthic. . ."

Figure 2: The images may be oriented and organized in a way that makes comparison easier and consumes less space. Figure 8: upper quartile boxes of F-1 and F-2 are

flawed.

---

## Referee Comment (RC2) · Anonymous Referee #2 · 9 Sep 2020

The paper "Vertical distribution of planktic foraminifera through an Oxygen Minimum Zone: how assemblages and shell morphology reflect oxygen concentrations" by Davis et al. presents an interesting study on planktic foraminifers from the Californian OMZ. The vertical distribution of planktic foraminifera seems to be controlled by oxygen availability and the porosity of G. hexagonus appears to be a phenotypic plastic trait that has the potential to be used as a paleoproxy for oxygen concentrations in the water column. While a lot of previous studies focused on the influence of oxygen availability on the distribution of benthic foraminifera, studies on the influence oxygen on planktic

foraminifera are scarce. The paper is well written and the results of this study are new and definitely worth being published in Biogeosciences. Nevertheless, there are a few points of moderate revision that should be addressed before publication:

The morphometric analyses were mainly done, using a normal light microscope. This has the advantage, that the use of the phenotypic plastic traits as paleoproxies can be done without electron-microscopy, which is both cheaper and faster. Unfortunately, this approach has also strong limitations. The pores of foraminifera are typically too small to measure their size correctly under a light microscope. Therefore, the measured total porosity is also likely to be very inaccurate. This might be one reason for the strong scattering in figure 9a and the low R2 of the correlation between porosity and dissolved oxygen. Regarding this correlation: What kind of fit has been used to determine R2 and P. Was it a linear fit? Is it possible to give the equation of the fit in the paper? I would recommend determining the porosity on a few electron microscope pictures and correlating them with the porosity measured on the same individuals using the light microscope method. This would give a coarse estimate about the accuracy of this method.

The authors have already done a similar comparison using micro-CT. Nevertheless, micro-CT is also at the limit of resolution, considering the size of pores in foraminiferal tests. In this context the authors state: "A comparison of the two methods carried out on a subset of shells (n = 31) showed that the results from the two approaches are correlated (R2 = 0.37, p-value < 0.001; Fig. 7), indicating that the less labor-intensive use of light-microscope measurements captures some of the same trend as the CT-based approach." In my opinion a correlation coefficient of 0.37 is too low, to make such a statement, when comparing the two methods.

G. hexagonus and H. parapelagica seem to be well adapted to oxygen depleted environments. This is a very interesting finding for planktic foraminifers. What I miss in this paper is a small discussion about different survival strategies of (benthic) foraminifera to oxygen depletion. They might apply to planktic foraminifera, too.

The finding that the size G. hexagonus specimens increases with decreasing oxygen is counterintuitive but very intriguing. A similar observation has been done on benthic foraminifera from the same region (Keating-Bitonti and Payne, 2017). The paper is already cited but it might be worth to mention the finding from above in the discussion. The porosity of G. hexagonus seems to decrease with ontogeny. This seems to be the opposite trend than in benthic foraminifera. As far as I know, usually the last chamber is the most porous. The ontogenetic trend might be a problem for the application as a paleoproxy. Could the authors think about a method, that minimizes the influence of this ontogenetic trend?

Line 439: "is consistent with a reduction of overall calcification in low oxygen, DIC rich environments, where precipitation and maintenance of a shell may be more metabolically expensive." Is there a reference that calcite precipitation is metabolically more expensive under oxygen depletion? Otherwise, this is very speculative. The formation of biomass, for example, is energetically favorable under oxygen depletion.

Line 426: The authors write that nitrate increased with depth. Is there a correlation between shell size and nitrate availability? In this case, the increased porosity might be just a secondary feature, due to the lower surface to volume ratio in larger individuals.

The following paper might be worth to be considered in the discussion: "Richirt, J., Champmartin, S., Schweizer, M. et al. Scaling laws explain foraminiferal pore patterns. Sci Rep 9, 9149 (2019). https://doi.org/10.1038/s41598-019-45617-x"

I think the pictures of G. hexagonus in figure 2 are not very representative of this species. Is it possible to add electron-micrographs of the two species from this figure, focusing on the morphological traits that characterize these species?

Other than these few moderate points of critique, this is a very intriguing, explorative pilot study on the development of a potential proxy for oxygen depletion in the water column and I would be happy to see this published in revised form.

---

## Author Comment (AC1) · 5 Oct 2020

Many thanks for your detailed comments on our manuscript. Each point is addressed individually below.

"Most importantly, I would strongly suggest to convey analyses of the molecular genetics of the specimens of the two morphotypes (line 394). For unequivocal proof of the species concept, modern papers of the kind presented here may accept the great opportunity of molecular genetics, and not only rely on the morphospecies concept. Also

line 404-406: "The shells of G. hexagonus in deeper, less oxygenated waters appeared more porous, larger, and less compact than those from shallower, more oxygenated environments." These specimens may or may not represent two different genotypes" We agree that it is possible there are more than one genotype and would love to have genetic data to include. Unfortunately, the preservation of these samples makes them unsuitable for genetic analyses. We also see molecular genetics as an important future step to better understanding these OMZ-affiliated taxa and hope this can be carried out in the coming years.

"Second most importantly, I would strongly suggest to change the statements in lines 375-379, which are not substantiated by data: "We hypothesize that G. hexagonus occupies low-oxygen mid-waters globally (i.e., in the Atlantic as well as the Indo-Pacific), but that its deep habitat, low abundance, and the historical dearth of surveys of living planktic foraminifera in low O2 regions along the western African margin have biased observations of G. hexagonus in the modern Atlantic." – Many studies of sediment trap and net tow samples in the South Atlantic off Namibia (Loncaric and colleagues from Bremen) and the Congo River mouth (Ufkes et al.), as well as surface sediment, would have certainly detected G. hexagonus if present. I have myself seen many net tow, sediment trap, and bottom sediment samples from the Atlantic and other ocean basins, including my PhD project on benthic foraminifers from surface sediments in the Gulf of Guinea, and I have never seen a test of hexagonus in that region." We will remove reference to the African margin, as we agree that we lack affirmative reporting of G. hexagonus in these regions in particular. However, there are reports from Atlantic sediment traps (Smart et al., 2018 as cited) as well as from recent sediments (e.g., the Brown Foraminiferal Database) which demonstrate that G. hexagonus is and has been present in the recent past in some regions of the Atlantic.

The following references have all been added as suggested "Line 40: please refer to the nice paper of Schmidtko et al. 2017 on modern OMZs" "Line 88: see also: Glock, N., et al. (2018) Nature Communications, 9, 1217, doi:10.1038/s41467-018-03647-5

Glock, N., et al. (2019) PNAS, 116 (8), 2860-2865, doi:10.1073/pnas.1813887116"
"Line 327: Please refer to Schiebel et al. (2004) for T. sacculifer in another study in a
region with a prominent subsurface OMZ, i.e. the Arabian Sea." "Line 369: refer again
to Schiebel et al. (2004)"

"Line 75: please add also Warren 1994, see Schiebel and Hemleben 2017" The refer-
ence from Schiebel and Hemleben (2017) appears to be "Warren, B.A. Context of the
suboxic layer in the Arabian Sea. Proc. Indian Acad. Sci. (Earth Planet Sci.) 103,
301–314 (1994). https://doi.org/10.1007/BF02839540". However, this article does not
mention foraminifera or plankton tows as relevant to Line 75. Is it possible that there is
another citation that should be added here?

"Line 125: please give the net strata depths and volume filtered for each net in Wishner
et al. 2019, 2020b here as well. This does not consume much space, and saves the
reader from consulting for additional literature." These can be found in the supplement,
reference to which will be included here.

"Lines 125-126: what was the filtered water volume?" This varied between tows and
can be found in the supplement, which is now referenced here.

"Line 227: Foraminifers do not really die in most cases, but reproduce. Therefore,
you may change "dead" for "empty" (tests)." This is true in most cases and will be
altered elsewhere in the manuscript. However, here we specifically reference a few
individuals found with cytoplasm (not empty) well below their photic zone habitat. This
is suggestive that they may not have successfully reproduced.

"Lines 299-301: "This species can be considered an indicator of an OMZ habitat and
may be useful as an OMZ marker in sedimentary records, as discussed below." This
should possibly be the final statement of the section. BTW: This finding is not new to
science; please refer to the respective literature." The sentence will be moved, and
while the finding of G. hexagonus associated with the OMZ is not new, it has not to the
best of our knowledge been used as an indicator of an overlying OMZ in the published

literature.

"Lines 302- 310 present a repetition of the "Results". Please rewrite." These lines will be rewritten with a greater emphasis on the implications rather than numeric abundances.

"Line 318: "larger" reads better than "more large" to me." There is a subtle but real distinction here. We don't have data on the distribution of size ("larger") but are rather arguing that more individual foraminifera in the relatively large size class (> 222 $\mu$m) sampled by our nets were present. We will rephrase to clarify.

"Lines 322-324: "Use of presence/absence of cytoplasm as an indicator for living foraminifera results in an overestimation of live individuals, as dead individuals may retain some cytoplasm while live individuals cannot be devoid of cytoplasm." This is possibly not entirely true, since decease is most often caused by reproduction, and cytoplasm is consumed and partly converted into offspring (see above)." We agree and this will be rephrased as "dead or post-reproductive individuals" as both cases can result in a shell retaining some cytoplasm.

"Line 427: Buchmann year of publication" I'm afraid, I don't understand this comment. Would it be possible for the reviewer to clarify so that we can address this?

"Lines 431-432: "(e.g., Bijma. . ." Some species increase in weight, others decrease. Please see Beer et al., Geology, 2010, using samples from the Arabian Sea, i.e. another OMZ region." This is an important point, but I would argue not directly relevant to the argument being made in the manuscript. Beer and other authors (for example also Weinkouf et al., 2016) raise the important caveat that carbonate chemistry may not be a primary driver of SNW in all species and regions, pointing to other drivers such as nutrients and temperature. However, it has not been shown, including by Beer et al., 2010, that any planktic foraminifer increases in weight as a response to decreasing carbonate chemistry. Our results are broadly consistent with a widely recognized carbonate ion driver in direction, though we do not explicitly discount other drivers, and

given the lack of both in situ carbonate chemistry and other (e.g., nutrient measurements) a further discussion of potential drivers of SNW is really outside of the scope of this manuscript.

"Line 447: better use outnumber or surpass instead of overwhelm." We will remove this descriptive language so that the phrase reads "However, the increase in size with decreased oxygen availability is such that. . ."

"Figure 2: The images may be oriented and organized in a way that makes comparison easier and consumes less space." This figure will be made more compact.

The following changes will be made as suggested: "Line 158: you may want to explain the abbreviations F and F-1 at first mention. Lines 212-214, and 220-221: You may skip the sentence on which species were present and absent. Many other species were possibly absent as well, and are not mentioned. The information on the presence and absence of species should also be available from figures and data tables. "Line 108: change reches to reaches" "Line 234: "Empty test assemblages" "Line 453: better "as in some benthic. . ."" "Figure 8: upper quartile boxes of F-1 and F-2 are flawed."

References Weinkauf, M. F. G., Kunze, J. G., Waniek, J. J., and Kučera, M., 2016, Seasonal Variation in Shell Calcification of Planktonic Foraminifera in the NE Atlantic Reveals Species-Specific Response to Temperature, Productivity, and Optimum Growth Conditions: PLOS ONE, v. 11, no. 2, p. e0148363.

―――――――――――――――――――

---

## Author Comment (AC2) · 5 Oct 2020

Many thanks for your thorough review and perspective on our manuscript. Each point is addressed individually below.

"The morphometric analyses were mainly done, using a normal light microscope. This has the advantage, that the use of the phenotypic plastic traits as paleoproxies can be done without electron-microscopy, which is both cheaper and faster. Unfortunately, this approach has also strong limitations. The pores of foraminifera are typically too small

[Figure]

to measure their size correctly under a light microscope. Therefore, the measured total porosity is also likely to be very inaccurate. This might be one reason for the strong scattering in figure 9a and the low R2 of the correlation between porosity and dissolved oxygen. . .. I would recommend determining the porosity on a few electron microscope pictures and correlating them with the porosity measured on the same individuals using the light microscope method. This would give a coarse estimate about the accuracy of this method. The authors have already done a similar comparison using micro-CT. Nevertheless, micro-CT is also at the limit of resolution, considering the size of pores in foraminiferal tests." Âň We are in full agreement with the reviewer here as to both the limitations and benefits of using light microscopy for measurements of porosity. We will add a line explicitly discussing the trade-offs in using this method in our revised draft. In the case of G. hexagonus, the pores are quite large (see images with more to be included), meaning that the greatest limitation in practice has been the curvature of the shells. Images from micro-CT scans were an excellent way to minimize this problem, by generating an essentially unlimited number of angles available from which to measure porosity. We have included additional CT-scan generated images demonstrating both this approach as well as the resolution of the method. In this particular case, SEM has the disadvantage of being a functionally destructive analysis. This is due to the need to mount quite fragile shells on carbon tape, coat them, and in some cases amputate chambers (to look at internal porosity). Given the limited number of shells available, and the limited benefits of SEM in addition to CT analyses, we have opted not to image our shells in that way. We leave open the possibility that this may be a preferable method for the development of a quantitative porosity-based proxy for oxygenation in G. hexagonus, but such a quantitative assessment is beyond the scope of the current paper, and likely this sample set. At this point, we are only able to demonstrate an empirical trend, captured by two very different approaches.

"Regarding this correlation: What kind of fit has been used to determine R2 and P. Was it a linear fit? Is it possible to give the equation of the fit in the paper?" "In this context the authors state: "A comparison of the two methods carried out on a subset of shells

(n = 31) showed that the results from the two approaches are correlated (R2 = 0.37, p-value < 0.001; Fig. 7), indicating that the less labor-intensive use of light-microscope measurements captures some of the same trend as the CTbased approach." In my opinion a correlation coefficient of 0.37 is too low, to make such a statement, when comparing the two methods". This is a linear fit and the equation will be included (y= x*0.23 + 2.64). The comparison is simply meant to demonstrate that multiple methods result in the same trend, but we would caution that due to the low predictive power of this relationship; light microscopy should not be used to estimate CT-based porosity. The two methods are not interchangeable, though both capture the same trend. We will make sure to clarify in our added discussion of the trade offs between imaging methodologies.

"G. hexagonus and H. parapelagica seem to be well adapted to oxygen depleted environments. This is a very interesting finding for planktic foraminifers. What I miss in this paper is a small discussion about different survival strategies of (benthic) foraminifera to oxygen depletion. They might apply to planktic foraminifera, too." We will add an additional short discussion of denitrification, dormancy, and kleptopasty in benthic foraminifera.

"The finding that the size G. hexagonus specimens increases with decreasing oxygen is counterintuitive but very intriguing. A similar observation has been done on benthic foraminifera from the same region (Keating-Bitonti and Payne, 2017). The paper is already cited but it might be worth to mention the finding from above in the discussion." We will elaborate to include a clause on their results as to size, in particular that only two of four species analyzed demonstrate the expected decrease in size with decreasing oxygen concentrations.

"The porosity of G. hexagonus seems to decrease with ontogeny. This seems to be the opposite trend than in benthic foraminifera. As far as I know, usually the last chamber is the most porous. The ontogenetic trend might be a problem for the application as a paleoproxy. Could the authors think about a method, that minimizes the influence of

this ontogenetic trend?" The reviewer raises a very good point. Unfortunately, all of these analyses were carried out on plankton tow specimens which may have been at different stages of (late) ontogeny, which complicates the meaning of a "last" chamber. It will probably be necessary for more work to be done on shells at their terminal stage, before the findings presented here could be presented as a quantitative paleoproxy.

"Line 439: "is consistent with a reduction of overall calcification in low oxygen, DIC rich environments, where precipitation and maintenance of a shell may be more metabolically expensive." Is there a reference that calcite precipitation is metabolically more expensive under oxygen depletion? Otherwise, this is very speculative. The formation of biomass, for example, is energetically favorable under oxygen depletion." The implication is not meant to be that calcite precipitation is metabolically more expensive under oxygen depletion (this may or may not be the case, but we whole-heartedly agree that there is currently no evidence), but under a DIC-rich environment coincident with the OMZ. This will be rephrased as "....reduction of overall calcification in low calcite saturation states associated with the OMZ...."

"Line 426: The authors write that nitrate increased with depth. Is there a correlation between shell size and nitrate availability? In this case, the increased porosity might be just a secondary feature, due to the lower surface to volume ratio in larger individuals." Nitrate availability increases in the region with depth as does size, however as we do not have data from the same tows, attempting to correlate these two parameters directly would probably be overreach. That increased porosity could in a sense be compensating for lower surface/volume ratio is an interesting possibility and will be added to the discussion.

"The following paper might be worth to be considered in the discussion: "Richirt, J., Champmartin, S., Schweizer, M. et al. Scaling laws explain foraminiferal pore patterns. Sci Rep 9, 9149 (2019). https://doi.org/10.1038/s41598-019-45617-x"" The findings of this paper will be referenced.

"I think the pictures of G. hexagonus in figure 2 are not very representative of this species. Is it possible to add electron-micrographs of the two species from this figure, focusing on the morphological traits that characterize these species?" We will add additional CT-scan images of G. hexagonus to a new panel in Figure 9, to demonstrate a greater range of the morphologies observed.

―――――――――――――――――――――――

---

## Author Response (AR2)

Reviewer 1:

*"About the comparison between CT and light microscopy: The authors now included the equation of the linear regression. For the sake of completeness it would be great if the authors can add the statistical errors of the (i.e. the error of the slope and the intercept)."*

The standard error has been added to the reported equation as: $y = 0.23(\pm 0.05) x + 2.64(\pm 1.49)$

*"One comment about the following sentence of the authors within the response letter:*
*"The reviewer raises a very good point. Unfortunately, all of these analyses were carried out on plankton tow specimens which may have been at different stages of (late) ontogeny, which complicates the meaning of a "last" chamber." In this case, using the term "last" chamber without further explanation might be misleading. Please clarify directly, that not all individuals were in the same ontogenetic stage."*

Thank you for pointing out that this needs further clarification in the manuscript. We have added the following to the methods: "We chose to focus our analysis of porosity on the most recent chamber as it was the chamber most likely to have formed under the conditions recorded at collection, however as the foraminifera analyzed had not yet reproduced, it is not possible to know whether this chamber would have been the terminal chamber, analogous to the final chamber in a fossil shell."

Ralf Schiebel

*"I understand that new data on the molecular genetics of G. hexagonus are not easy to get on short notice. I also understand that Davis and coauthors may have the feeling that G. hexagonus is a normal component of Atlantic assemblages of modern planktic foraminifers. However, I know the specimens presented by Smart et al. (2018) by heart; the specimens really look like G. hexagonus, but they are not numerous. Therefore, we could not get data on the genotypes. Also, the G. hexagonus found in sediment assemblages are not numerous, and may have been transported into the Atlantic Ocean by currents (if these are G. hexagonus at all; I did not have a look at them). Therefore, I would suggest to be careful with stating that G. hexagonus is present in all modern oceans; this would need proof of an entire life cycle of G. hexagonus, including reproduction, in Atlantic water masses. Having said all this, Section "4.2 Globorotaloides hexagonus as an OMZ Indicator Species" reads better than before. The authors may still add that "final proof of the presence of G. hexagonus in the Atlantic may be provided by molecular genetics", which I would regard as a good way to solve the issue."*

We appreciate these comments and the reviewers experience in this area. However, we would point out that G. hexagonus is a rare taxa globally, making up generally <3% of the sedimentary or sediment trap assemblage even in the Indo-Pacific. Nor has the entire life cycle been demonstrated anywhere in the modern ocean. We understand that suggesting that hexagonus may be present in the modern Atlantic goes against some received wisdom, but given that it appears not to be entirely absent, we believe that it is worthwhile to revisit this question as we seek to better understand the ecology and biogeography of G. hexagonus. Ultimately, we are in agreement that molecular genetics could be useful in answering this question and have included the suggested addition rephrased as "However, additional evidence, such as molecular genetics, may be required to finally resolve this question."

Minor points:
*"Line 44: Breitburg et al., 2108 should possibly read 2018"*
Of course it should! Thank you.

*"Line 196: "extremely oxygen depleted OMZ" is not a term defined below in this paper. I would suggest to drop "extremely". Along the same lines, the paper would benefit from dropping a number of adjectives in places, which would make reading more scientific."*
"Extremely" has been dropped and a few additional adjectives have been removed throughout where not entirely justified.

*"Line 211: change "transitional (nets sampling between these two extremes)" to "between these two concentrations"."*
This has been altered as suggested.

*"Line 213: I would suggest to change "the densest population" to „highest standing stock", which is possibly more correct in a scientific way. In general, densities may be changed for standing stock throughout the paper."*
This has been changed as suggested

*"Lines 244-252: I would suggest to add the numbers of empty tests to the Table 1."*
These have been added.

*"Line 273: better change "interaction" to "relationship"; interaction insinuates determination, which is possibly not the case."*
This has been changed

*"Line 287: "Globorotaloides hexagonus tests were light for their size", relates to something that is not mentioned. "Light" compared to what?"*
This has been removed.

*"Line 299: Please briefly describe what is meant by "compactness" and "aspect ratio"."*
These two parameters are defined on lines 178-183 as "The compactness of tests was assessed as the ratio of the 2-dimensional surface area to the area of a circle (the most compact possible geometry) of the same perimeter. The aspect ratio was defined as the ratio between the height (longest dimension) and width (perpendicular to the longest dimension) as measured in the AutoMorph software (Hsiang et al., 2016)."

*"Line 337-339: „It is more likely that cytoplasm-bearing tests of T. sacculifer found below the photic zone are a consequence of their very high abundance in the surface ocean and reflected premature mortality and/or the retention of cytoplasm following reproduction." Did these specimens still have their spines? If yes, they may not have reproduced."*

This is a wonderful though. For the most part, no, but spines were not consistently preserved in these tows, even in near-surface cytoplasm-bearing shells. It is possible this is the result of relatively long tow times (~1 hour in most cases), but regardless of cause poor spine preservation would not be a reliably indicative of reproduction in these particular samples.

*"Line 358: Predation may be another reason to be taken into account."*
Great point. This has been added

*"Line 427 of the 1st version of manuscript: Sorry, this should have read Buchwald. Lines 441-442 in the new version of manuscript does still contain the same transposed digits: "Buchwald et al., 2105", which may be changed to 2015."*
Thank you for catching this. It has been corrected

*"Lines 456-464 presents a repetition of statements presented above, and may be rewritten or removed."*
We've removed some repetitive language but retained the discussion of increased chamber number.

*"Final paragraph, lines 465-480: Another explanation for larger test sizes may be that G. hexagonus continuously grew larger under less optimal environmental conditions, i.e., lack of oxygen, and only reproduced with a delay when the environmental conditions had improved to support survival of the offspring (see Mojtahid et al., 2015 for G. ruber) and Schiebel and Hemleben (2017) for a more general explanation of the phenomenon."*
We agree that delayed or slower reproduction at lower oxygen could be a plausible, however, it is unlikely that environmental conditions will "improve" in this habitat. i.e. the deep > 400 m OMZ where the largest individuals tend to be found is unlikely to ever be truly released from oxygen pressure. We have added this general idea to these lines as "Larger sizes could also result from delayed reproduction at lower oxygen levels."

*"Figure 1: O2 concentration larger than 7.5 ml L-2 are not presented in the map, and may be cut from the scale. By doing so, the relevant part of the scale may be more detailed."*
Figure 1 has been redrawn and rescaled.

*"Figure 9: The few red and yellow markers are not easy to see in the paper copy of the figure, and I would suggest to use darker color. Also, the green and blue markers would benefit from darker color."*
We have replaced these markers with darker hues of the same colors.